# Type-Aware Decomposed Framework
# for Few-Shot Named Entity Recognition

**Yongqi Li[1], Yu Yu[1], Tieyun Qian[1,2,*]**

[1] School of Computer Science, Wuhan University, China

[2] Intellectual Computing Laboratory for Cultural Heritage, Wuhan University, China

{liyongqi,Yu.Yu1024,qty}@whu.edu.cn

## Abstract

Despite the recent success achieved by several two-stage prototypical networks in few-shot named entity recognition (NER) task, *the over-detected false spans* at the span detection stage and *the inaccurate and unstable prototypes* at the type classification stage remain to be challenging problems. In this paper, we propose a novel **T**ype-**A**ware **D**ecomposed framework, namely TadNER, to solve these problems. We first present *a type-aware span filtering strategy* to filter out false spans by removing those semantically far away from type names. We then present *a type-aware contrastive learning strategy* to construct more accurate and stable prototypes by jointly exploiting support samples and type names as references. Extensive experiments on various benchmarks prove that our proposed TadNER framework yields a new state-of-the-art performance. [1]

## 1 Introduction

Named entity recognition (NER) aims to detect entity spans and classify them into pre-defined categories (entity types). When there are sufficient labeled data, deep learning-based methods (Huang et al., 2015; Ma and Hovy, 2016; Lample et al., 2016; Chiu and Nichols, 2016) can get impressive performance. In real applications, it is desirable to recognize new categories which are unseen in training/source domain. However, collecting extra labeled data for these new types will be surely time-consuming and labour-expensive. Consequently, few-shot NER (Fritzler et al., 2019; Yang and Katiyar, 2020), which involves identifying unseen entity types based on a few labeled samples for each class (i.e., *support samples*) in test domain, has attracted great research interests in recent years.

End-to-end metric learning based methods (Yang and Katiyar, 2020; Das et al., 2022) are the main-

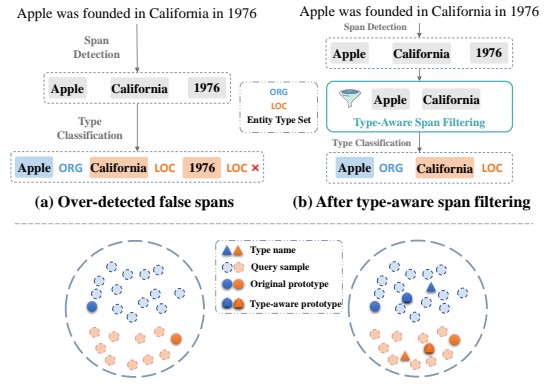

Figure 1: (a) shows over-detected false spans, (b) shows spans got by adopting our type-aware span filtering strategy. (c) shows inaccurate and unstable prototypes, (d) shows prototypes got by adopting our type-aware contrastive learning strategy.

stream in few-shot NER. These methods need to simultaneously learn the complex structure consisting of entity boundary and entity type. When the domain gap is large, their performance will drop dramatically because it is extremely hard to capture such complicated structure information with only a few support examples for domain adaptation. This leads to the insufficient learning of boundary information, resulting that these methods often misclassify entity boundaries and cannot obtain very satisfying performance.

Recently, there is an emerging trend in adopting two-stage prototypical networks (Wang et al., 2022; Ma et al., 2022c) for few-shot NER, which decompose NER into two separate *span extraction* and *type classification* tasks and perform one task at each stage. Since decomposed methods only need to handle one single boundary detection task at the first stage, they can find more accurate boundaries and obtain better performance than end-to-end approaches.

While making good progress, these two-stage prototypical networks still face two challenging problems, i.e., *the over-detected false spans* and *the*

---

[*] Corresponding author.

[1] Our code and data will be available at https://github.com/NLPWM-WHU/TadNER.

*inaccurate and unstable prototypes* in corresponding stages. (1) At the span extraction stage in test phase, the decomposed approaches usually recall many over-detected false spans whose types only exist in the source domain. For example, "1976" in Figure 1 (a) belongs to a DATE type in the source domain since there are many samples like "Obama was born in 1961" in training, and thus it is easily recognized as a span by the span detector. However, there is no such label in the test domain and "1976" is thus assigned a false LOC type. (2) The prototypical networks in decomposed methods target at learning a type-agnostic metric similarity function to classify entities in test samples (*i.e., query samples*) via their distance to prototypes. Since the prototypes are constructed using very few support samples in the type-agnostic feature space, they might be inaccurate and unstable. For example, in Figure 1 (c), a prototype is just the support sample in one-shot NER and thus deviates far away from the real class center.

Based on the above observations, we propose a **T**ype-**A**ware **D**ecomposed framework, namely TadNER, for few-shot **NER**. Our method follows the span detection and type classification learning scheme in the decomposed framework but moves two steps further to overcome the aforementioned issues.

Firstly, we present *a type-aware span filtering strategy* to filter out false spans by removing those semantically far away from type names [2]. By this means, the over-detected spans like "1976" whose types do not exist in test domain can be removed due to the long semantic distance to type names, as shown in Figure 1 (b).

Secondly, we present *a type-aware contrastive learning strategy* to construct more accurate and stable prototypes by jointly leveraging type names and support samples as references. Through this way, the type names can serve as the guidance for prototypes and make them not deviate too far away from the class centers even in some extreme outlier cases, as shown in Figure 1 (d).

Extensive experimental results on 5 benchmark datasets demonstrate the superiority of our TadNER over the state-of-the-art decomposed methods. In particular, in the hard intra Few-NERD and 1-shot Domain Transfer settings, TadNER achieves a 8% and 9% absolute F1 increase, respectively.

---

[2]Note that though type assignments are unknown in few-shot NER, the type names (labels) in test domain are provided.

## 2 Method

In this section, we formally present our proposed TadNER. The overall structure of our TadNER is shown in Figure 2. Note that the type-aware contrastive learning and type-aware span filtering strategies take effect at the type classification stage in the training and test domain, respectively.

**Task Formulation** Given a sequence $X = \{x_1, x_2, ..., x_N\}$ with $N$ tokens, NER aims to assign each token $x_i$ a corresponding label $y_i \in \mathcal{T} \cup \{O\}$, where $\mathcal{T}$ is the entity type set and O denotes the non-entity label. For few-shot NER, a model is trained in a source domain dataset $\mathcal{D}_{source}$ with the entity type set $\mathcal{T}_{source} = \{t_1, t_2, ...t_m\}$. The model is then fine-tuned in a test/target domain dataset $\mathcal{D}_{target}$ with the entity type set $\mathcal{T}_{target} = \{t_1, t_2, ...t_n\}$ using a given support set $\mathcal{S}_{target}$. The entity token set and corresponding label set in $\mathcal{S}_{target}$ are denoted as $E^s = \{e_1^s, e_2^s, ..., e_M^s\}$ and $Y^s = \{y_1^s, y_2^s, ..., y_M^s\}$, where $y_i^s \in \mathcal{T}_{target}$ is the label and $M$ is the number of entity tokens. The model is supposed to recognize entities in the query set $\mathcal{Q}_{target}$ of the target domain. Besides, $\mathcal{T}_{source}$ and $\mathcal{T}_{target}$ have no or very little overlap, making few-shot NER very challenging. More specifically, in the $n$-way $k$-shot setting, there are $n$ labels in $\mathcal{T}_{target}$ and $k$ examples associated with each label in the support set $\mathcal{S}_{target}$.

### 2.1 Source Domain Training

The source domain training consists of span detection and type classification stages. The procedure is shown in Figure 2 (a).

#### 2.1.1 Span Detection

The span detection stage is formulated as a sequence labeling task, similar to an existing decomposed NER model (Ma et al., 2022c). We adopt BERT (Devlin et al., 2019) with parameters $\theta_1$ as the PLM encoder $f_{\theta_1}$. Given an input sentence $X = \{x_1, x_2, ..., x_N\}$, the encoder produces contextualized representations for each token as:

$$\mathbf{H} = [\mathbf{h_1}, ..., \mathbf{h_N}] = f_{\theta_1}([x_1, ..., x_N]), \quad (1)$$

where $\mathbf{H} \in \mathbb{R}^{N*r}$ [3]. $\mathbf{H}$ is then fed into a classification layer consisting of a dropout layer (Srivastava et al., 2014) and a linear layer to get the probability distribution $\mathbf{P} = [\mathbf{p(x_1)}, ..., \mathbf{p(x_N)}]$ ($\mathbf{p(x_i)} \in \mathbb{R}^{|C|}$, $C = \{I, O\}$) [4] using a softmax

---

[3]In this paper, $r$ denotes the hidden size of the PLM.
[4]In Appendix A.6, we perform a detailed analysis using the IO, BIO, and BIOES tagging schemes.

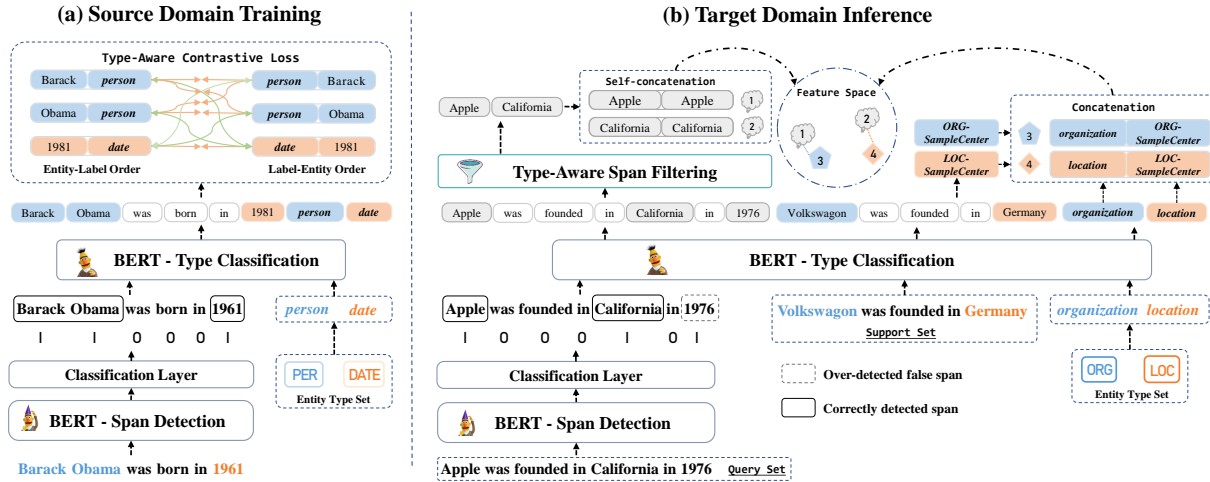

Figure 2: The overall structure of our proposed TadNER framework. (a) Training in the source domain. (b) Inference on the query set by utilizing the support samples in the target domain. Note that the source and target domains have different entity type sets.

function:

$$\mathbf{p}(\mathbf{x_i}) = softmax(Dropout(\mathbf{W} \cdot \mathbf{h_i} + \mathbf{b})), \quad (2)$$

where $\mathbf{W} \in \mathbb{R}^{|C|*r}$ and $\mathbf{b} \in \mathbb{R}^{|C|}$ are the weight matrix and bias.

After that, the training loss is formulated by the averaged cross-entropy of the probability distribution and the ground-truth labels:

$$\mathcal{L}_{span} = \frac{1}{N} \sum_{i=1}^{N} CrossEntropy(y_i, \mathbf{p}(\mathbf{x_i})), \quad (3)$$

where $y_i$=0 when the $i$-th token is O-token, $y_i$=1 otherwise. Specifically, we denote the training loss of span detection stage as $\mathcal{L}_{span}$. During the training procedure, the parameters $\{\theta_1, \mathbf{W}, \mathbf{b}\}$ are updated to minimize $\mathcal{L}_{span}$.

### 2.1.2 Type Classification

**Representation**  Given an input sentence $X$, we only select entity-tokens $E = \{e_1, e_2, ..., e_M\}$ ($E \subset X$) with ground-truth labels $Y = \{y_1, y_2, ..., y_M\}$ for the training of this stage. For the entity type set $\mathcal{T}_{source} = \{t_1, t_2, ..., t_m\}$ of the source domain $D_{source}$, we manually convert them into their corresponding type names $\mathcal{T}'_{source} = \mathrm{Map}(\mathcal{T}_{source}) = \{t'_1, t'_2, ..., t'_m\}$[5].

After that, to obtain tokens with type name information, which are further used for calculating contrastive loss, we concatenate entity tokens with their corresponding labels in two orders, i.e., entity-label order and label-entity order. Here we use

---

[5]Map() is the function used to convert a label to a type name, e.g. "PER" to "person". Please refer to Appendix A.7 for type names of all datasets.

another encoder $f_{\theta_2}$ with parameters $\theta_2$ to obtain contextual representations:

$$\mathbf{h_i^{el}} = f_{\theta_2}(e_i) \oplus f_{\theta_2}(\mathrm{Map}(y_i)) \quad (4)$$

$$\mathbf{h_i^{le}} = f_{\theta_2}(\mathrm{Map}(y_i)) \oplus f_{\theta_2}(e_i), \quad (5)$$

where $\oplus$ is the concatenation operator, and $\mathbf{h_i^{el}}$ and $\mathbf{h_i^{le}}$ denote two kinds of type-aware representations of the entity-token $e_i$, which are obtained in entity-label order and label-entity order, respectively.

**Type-Aware Contrastive Learning**  To learn a generalized and type-aware feature space, which can further be used for constructing more accurate and stable prototypes, we borrow the idea of contrastive learning (Khosla et al., 2020) and use two kinds of type-aware token representations mentioned above to construct positive and negative pairs as shown in Figure 2 (a), i.e., those with the same label in different orders as positive pairs and those with different labels as negative pairs. The type-aware contrastive loss is calculated as:

$$\mathcal{L}_{type} = -\sum_{i=1}^{M} \log \frac{\frac{1}{\|Z_i\|} \sum_{z \in Z_i} \exp(sim(\mathbf{h_i^{el}}, \mathbf{h_z^{le}})/\tau)}{\sum_{j=1}^{M} \exp(sim(\mathbf{h_i^{el}}, \mathbf{h_j^{le}})/\tau)}, \quad (6)$$

$$Z_i = \{z \mid 1 \le z \le M, y_z = y_i\}, \quad (7)$$

$$sim(\mathbf{h_i^{el}}, \mathbf{h_j^{le}}) = \frac{\mathbf{h_i^{el}} \cdot \mathbf{h_j^{le\top}}}{\sum_{k=1}^{M} (\mathbf{h_k^{el}} \cdot \mathbf{h_j^{le\top}})}, \quad (8)$$

where $M$ is the number of entity tokens in a batch and $Z_i$ is the set of positive samples with the same

label type $y_i$. Here we adopt the dot product with a normalization factor as the similarity function $sim()$. We also add a temperature hyper-parameter $\tau$ for focusing more on difficult pairs (Chen et al., 2020). During the source domain training, the parameters $\theta_2$ are updated to minimize $\mathcal{L}_{type}$.

## 2.2 Target Domain Inference

As illustrated in Figure 2 (b), during the target domain inference, we first extract candidate spans in query sentences and then remove over-detected false spans via the type-aware span filtering strategy. Finally, we classify remaining candidate spans into certain entity types to get the final results.

### 2.2.1 Span Detection

The span detector with its parameters $\{\theta_1, \mathbf{W}, \mathbf{b}\}$ trained in the source domain is further fine-tuned with samples in the support set $\mathcal{S}_{target}$ in the target domain to minimize $\mathcal{L}_{span}$ in Eq.(3). To alleviate the risk of over-fitting, we adopt a loss-based early stopping strategy, i.e., stopping the fine-tuning procedure once the loss rises $\beta$ times continuously, where $\beta$ is a hyper-parameter.

After fine-tuning the span detector, we use it to detect entity words of query sentences in $\mathcal{Q}_{target}$ and then consider continuous entity words as a candidate span, e.g., "Barack Obama". Finally, we obtain the candidate span set $C_{span}$ containing all candidate spans, which will be assigned entity types at the type classification stage.

### 2.2.2 Type Classification

**Domain Adaption** Benefiting from the generalized and type-aware feature space trained in the source domain, we can further get a domain-specific encoder $f_{\theta_2'}$ via fine-tuning with the following loss:

$$\mathcal{L}_{label} = \frac{1}{M} \sum_{i=1}^{M} \frac{s(e_i^s, \mathtt{Map}(y_i^s))}{\sum\limits_{t_j \in \mathcal{T}_{target}} s(e_i^s, \mathtt{Map}(t_j))}, \qquad (9)$$

$$s(p, q) = f_{\theta_2}(p) \cdot f_{\theta_2}(q)^{\top}. \qquad (10)$$

**Type-Aware Span Filtering** As we illustrate in the introduction, the span detector may generate some over-detected false spans whose type names only belong the source domain, since the semantics of entity type names are not considered at the span detection stage. To solve this problem, we propose a type-aware span filtering strategy during the inference phase to remove these false spans. Intuitively, the semantic distance of these false spans

is far from all the golden type names. Based on this assumption, we calculate a threshold $\gamma_t$ with the fine-tuned encoder $f_{\theta_2'}$ using entity tokens and corresponding type names in the support set:

$$\gamma_t = \min_{1 \le i \le M} f_{\theta_2'}(e_i^s) \cdot f_{\theta_2'}(\mathtt{Map}(y_i^s))^{\top}. \qquad (11)$$

This threshold $\gamma_t$ is used to remove the over-detected false spans. And the remaining candidate spans will be assigned corresponding labels.

**Type-Aware Prototype Construction** We can construct a type-aware prototype for each entity type $t_j \in \mathcal{T}_{target}$, which is more accurate and stable owing to the generalized and type-aware feature space learned in the source domain:

$$\mathbf{p_j} = f_{\theta_2'}(\mathtt{Map}(t_j)) \oplus \frac{1}{\|Z_j\|} \sum_{i \in Z_j} f_{\theta_2'}(e_i^s), \qquad (12)$$

$$Z_j = \{i \,|\, 1 \le i \le M, y_i^s = t_j\}, \qquad (13)$$

where $\oplus$ is the concatenation operator and $Z_j$ denotes the set of entity words with the label type $t_j$ in the support set.

**Inference** For each remaining candidate span $s_i$, we assign it a label type $t_j \in \mathcal{T}_{target}$ with the highest similarity:

$$y_{pred} = \arg\max_{t_j, t_j \in \mathcal{T}_{target}} (\mathbf{h_i} \cdot \mathbf{p_j}^{\top}), \qquad (14)$$

$$\mathbf{h_i} = f_{\theta_2'}(s_i) \oplus f_{\theta_2'}(s_i), \qquad (15)$$

where $\mathbf{p_j}$ is the type-aware prototype representation corresponding to the label type $t_j$, and $y_{pred}$ is the predicted label type of the candidate span $s_i$. $\mathbf{h_i}$ is the self-concatenated representation of $s_i$ for consistency with the dimension of the prototype $\mathbf{p_j}$. The entire procedure of inference in the target domain is presented in Appendix A.1.

## 3 Experiments

### 3.1 Evaluation Protocal

**Datasets** Ding et al. (2021) propose a large scale dataset **Few-NERD** for few-shot NER, which contains 66 fine-grained entity types across 8 coarse-grained entity types. It contains intra and inter tasks where the train/dev/test sets are divided according to the coarse-grained and fine-grained types, respectively. Besides, following Das et al. (2022), we also conduct **Domain Transfer** experiments, where data are from different text domains

| Paradigms | Models | Intra | | | | | Inter | | | | |
|---|---|---|---|---|---|---|---|---|---|---|---|
| | | 1~2-shot | | 5~10-shot | | Avg. | 1~2-shot | | 5~10-shot | | Avg. |
| | | 5 way | 10 way | 5 way | 10 way | | 5 way | 10 way | 5 way | 10 way | |
| One-stage | ProtoBERT† | 20.76±0.84 | 15.05±0.44 | 42.54±0.94 | 35.40±0.13 | 28.44 | 38.83±1.49 | 32.45±0.79 | 58.79±0.44 | 52.92±0.37 | 45.75 |
| | NNShot† | 25.78±0.91 | 18.27±0.41 | 36.18±0.79 | 27.38±0.53 | 26.90 | 47.24±1.00 | 38.87±0.21 | 55.64±0.63 | 49.57±2.73 | 47.83 |
| | StructShot† | 30.21±0.90 | 21.03±1.13 | 38.00±1.29 | 26.42±0.60 | 28.92 | 51.88±0.69 | 43.34±0.10 | 57.32±0.63 | 49.57±3.08 | 50.53 |
| | FSLS* | 30.38±2.85 | 28.31±4.03 | 46.85±3.49 | 40.76±3.18 | 36.58 | 44.52±4.59 | 44.01±3.35 | 59.74±2.51 | 56.67±1.75 | 51.24 |
| | CONTaiNER* | 41.51±0.07 | 36.62±0.04 | 57.83±0.01 | 51.04±0.24 | 46.75 | 50.92±0.29 | 47.02±0.24 | 63.35±0.07 | 60.14±0.16 | 55.36 |
| Two-stage | ESD† | 36.08±1.60 | 30.00±0.70 | 52.14±1.50 | 42.15±2.60 | 40.09 | 59.29±1.25 | 52.16±0.79 | 69.06±0.80 | 64.00±0.43 | 61.13 |
| | DecomposedMetaNER† | 49.48±0.85 | 42.84±0.46 | 62.92±0.57 | 57.31±0.25 | 53.14 | 64.75±0.35 | 58.65±0.43 | 71.49±0.47 | 68.11±0.05 | 65.75 |
| | **TadNER** | **60.78±0.32** | **55.44±0.08** | **67.94±0.17** | **60.87±0.22** | **61.26** | **64.83±0.14** | **64.06±0.19** | **72.12±0.12** | **69.94±0.15** | **67.74** |

Table 1: F1 scores with standard deviations for Few-NERD. † denotes the results reported by Ma et al. (2022c). * denotes the results reported by our replication using data of the same version. The best results are in **bold** and the second best ones are underlined.

| Paradigms | Models | 1-shot | | | | | 5-shot | | | | |
|---|---|---|---|---|---|---|---|---|---|---|---|
| | | I2B2 | CoNLL | WNUT | GUM | Avg. | I2B2 | CoNLL | WNUT | GUM | Avg. |
| One-stage | ProtoBERT† | 13.4±3.0 | 49.9±8.6 | 17.4±4.9 | 17.8±3.5 | 24.6 | 17.9±1.8 | 61.3±9.1 | 22.8±4.5 | 19.5±3.4 | 30.4 |
| | NNShot† | 15.3±1.6 | 61.2±10.4 | 22.7±7.4 | 10.5±2.9 | 27.4 | 22.0±1.5 | 74.1±2.3 | 27.3±5.4 | 15.9±1.8 | 34.8 |
| | StructShot† | 21.4±3.8 | 62.4±10.5 | 24.2±8.0 | 7.8±2.1 | 29.0 | 30.3±2.1 | 74.8±2.4 | 30.4±6.5 | 13.3±1.3 | 37.2 |
| | FSLS* | 18.3±3.5 | 50.9±6.5 | 14.3±5.5 | 12.6±2.8 | 24.0 | 25.4±2.7 | 63.9±3.3 | 24.0±3.2 | 18.8 ±2.2 | 33.1 |
| | CONTaiNER† | 21.5±1.7 | 61.2±10.7 | 27.5±1.9 | 18.5±4.9 | 32.2 | 36.7±2.1 | 75.8±2.7 | 32.5±3.8 | 25.2±2.7 | 42.6 |
| Two-stage | DecomposedMetaNER* | 15.5±3.0 | 61.2±9.2 | 27.7±5.3 | 20.3±4.2 | 31.2 | 19.8±2.6 | 75.2±5.8 | 29.8±3.9 | 33.5±2.4 | 39.6 |
| | **TadNER** | **39.3±3.8** | **70.4±10.6** | **32.8±4.8** | **24.2±4.1** | **41.7** | **45.2±2.3** | **80.5±3.6** | **34.5±4.6** | **35.1±2.2** | **48.8** |

Table 2: F1 scores with standard deviations for Domain Transfer. † denotes the results reported by Das et al. (2022). * denotes the results reported by our replication. Since no previous two-stage methods have conducted experiments under this setting, we choose the strong DecomposedMetaNER for reproduction experiments, and * denotes the results reported by our replication. The best results are in **bold** and the second best ones are underlined.

(e.g., Wiki, News). We take OntoNotes (General) (Weischedel et al., 2013) as our source domain, and evaluate few-shot performances on I2B2 (Medical) (Stubbs and Uzuner, 2015), CoNLL (News) (Tjong Kim Sang and De Meulder, 2003), WNUT (Social) (Derczynski et al., 2017) and GUM (Zeldes, 2017) datasets.

**Baselines** We compare our proposed TadNER with many strong baselines, including *one-stage* and *two-stage* types. The *one-stage* baselines include ProtoBERT (Snell et al., 2017), NNShot (Yang and Katiyar, 2020), Struct-Shot (Yang and Katiyar, 2020), FSLS (Ma et al., 2022a) and CONTaiNER (Das et al., 2022). Note that FSLS also adopts type names. The *two-stage* baselines include ESD (Wang et al., 2022) and the DecomposedMetaNER (Ma et al., 2022c) [6].

## 3.2 Main Results

Table 1 and 2 report the comparison results between our method and baselines under Few-NERD [7] and Domain Transfer, respectively. We have the following important observations: 1) Our model demonstrates superiority under Few-NERD settings. Notably, in the more challenging intra task, our TadNER achieves an average 8.2% increase in F1 score. Besides, our model outperforms baselines by 10.5% and 9.2% under 1-shot and 5-shot Domain Transfer settings, respectively. 2) Particularly, when provided with very few samples (e.g., 1-shot), the improvements become even more significant, which is a very attractive property. 3) The performance of DecomposedMetaNER, a competing model, severely deteriorates under certain settings, such as I2B2. This is primarily due to the presence of numerous sentences without entities, leading to multiple false detected spans. In contrast, our TadNER effectively mitigates this issue through the type-aware span filtering strategy, successfully removing false spans and achieving promising results.

---

[6] Please refer to Appendix A.2-A.5 for more descriptions about datasets, evaluation methods, baselines and implementation details.

[7] Results are tested with the latest version of data from https://ningding97.github.io/fewnerd/, which is corresponding with https://github.com/microsoft/vert-papers/tree/master/papers/DecomposedMetaNER#few-nerd-arxiv-v6-version.

| **C1:** Query sentence: | with the promotion of emrespor to the turkish tff third league at the end of the 2011 season |
|---|---|
| DecomposedMetaNER: | **organization-sportsteam: emrespor** (√), **turkish tff third league** (✗) |
| **TadNER (ours):** | **organization-sportsteam: emrespor** (√) **organization-sportsleague: turkish tff third league** (√) |

| **C2:** Query sentence: | Leicestershire beat Somerset by an innings and 39 runs in two days. |
|---|---|
| DecomposedMetaNER: | **ORG: Leicestershire** (√) **LOC: Somerset** (✗), **two** (✗) |
| **TadNER (ours):** | **ORG: Leicestershire** (√), **Somerset** (√) |

Figure 3: Case study. C1 and C2 are from Few-NERD intra and CoNLL2003 in Cross datasets, respectively, and organization-sportsteam, organization-sportsleague, ORG and LOC are entity types.

## 3.3 Ablation Study

To validate the effectiveness of the main components in TadNER, we introduce the following variant baselines for the ablation study: 1) TadNER *w/o* Type-Aware Span Filtering (TASF) removes the type-aware span filtering strategy and directly feeds all spans detected at span detection stage to type classification. 2) TadNER *w/o* Type Names (TN) further replaces type names with random vectors when calculating contrastive loss and constructs class prototypes using only the support samples. 3) TadNER *w/o* Span-Finetune skips the target domain adaptation of the span detection stage. 4) TadNER *w/o* Type-Finetune skips the target domain adaptation of the type classification stage.

| **Models** | **1-shot** | | | | **5-shot** | | | | **Avg.** |
|---|---|---|---|---|---|---|---|---|---|
| | I2B2 | CoNLL | WNUT | GUM | I2B2 | CoNLL | WNUT | GUM | |
| **TadNER** | **39.3** | **70.4** | **32.8** | 24.2 | **45.2** | **80.5** | **34.5** | **35.1** | **45.3** |
| *w/o* TASF | 21.2 | 68.5 | 31.6 | **24.2** | 27.4 | 80.1 | 34.3 | **35.1** | 40.3 |
| *w/o* TN | 20.0 | 65.6 | 28.3 | 20.3 | 26.2 | 76.3 | 33.8 | 33.2 | 38.0 |
| *w/o* Span-Finetune | 37.0 | 52.5 | 30.7 | 15.0 | 40.1 | 50.8 | 31.7 | 16.2 | 34.3 |
| *w/o* Type-Finetune | 37.6 | 68.3 | 32.3 | 20.3 | 45.2 | 76.3 | 33.6 | 27.9 | 42.7 |

Table 3: Results (F1 scores) for ablation study under Domain Transfer settings. The best results are in **bold**.

From Table 3, we can observe that: 1) The removal of the type-aware span filtering strategy leads to a drop in performance across most cases, particularly in entity-sparse datasets like I2B2, where a large number of false positive spans are detected. Besides, for entity-dense datasets like GUM, the performance is not harmed by the span filtering strategy, which proves the robustness and effectiveness of our model in various real-world applications. 2) The omission of type names also results in a significant decrease in performance, indicating that our model indeed learns a type-aware feature space, which plays a crucial role in few-shot scenarios. 3) The elimination of finetuning in the span detection and type classification stages exhibits a substantial performance drop. This demon-

strates that the training objective in the source domain training phase aligns well with the target domain finetuning phase via task decomposition and contrastive learning strategy, despite having different entity classes. As a result, the model can effectively utilize the provided support samples from the target domain, enhancing its performance in few-shot scenarios.

## 3.4 Case Study

To examine how our model accurately constructs prototypes and filters out over-detected false spans with the help of type names, we randomly select one query sentence from Few-NERD intra and CoNLL2003 for case study. We compare TadNER with DecomposedMetaNER (Ma et al., 2022c), which also belongs to the two-stage methods.

As shown in Figure 3, in the first case, our model correctly predicts "turkish tff third league" as "organization-sportsleague" type, while DecomposedMetaNER identifies it as a wrong "organization-sportsteam" type. Since the type name and the entity span have an overlapping word "league", incorporating the type name into the construction of the prototype will make the identification much easier. Conversely, without the type name, it would be hard to distinguish two categories of entities because they both represent "sports-related organizations".

In the second case, DecomposedMetaNER incorrectly identifies "two" as an entity span and then assigns it a wrong entity type "LOC", since there are many samples like "The two sides had not met since Oct. 18" in the source domain Ontonotes, where "two" is an entity of "CARDINAL" type. In contrast, our TadNER successfully removes this false span via the type-aware span filtering strategy.

## 3.5 Impact of Type Names

To further explore the impact of incorporating the semantics of type names and whether model perfor-

mance is sensitive to these converted type names. We perform experiments with the following variants of type names: 1) Original type names, which are used in our main comparision experiments. 2) Synonymous type names. We generate three synonyms for each original type name as variants using ChatGPT. These synonyms were automatically generated to explore the effect of different but related type names on model performance. 3) Meaningless type names, e.g., "label 1" and "label 2". 4) Misleading type names, e.g., "person" for "LOC" and "location' for "PER" in the CoNLL dataset. Please refer to Appendix A.7 for details.

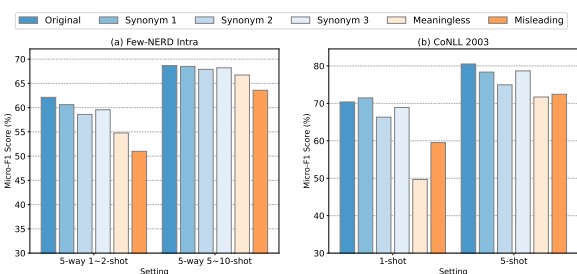

Figure 4: F1 Scores on Few-NERD Intra and CoNLL 2003 with different variants of type names.

As shown from the Figure 4, we can make the following observations: 1) All three variants of synonym type names have comparable performance, indicating that our method is robust to different ways of transforming type names. However, the best way is still the direct transformation method, such as "person" for "PER", which is how we obtain the original type names. 2) Irrelevant or incorrect information in meaningless and misleading type names leads to a significant degradation in model performance, indicating that the semantics associated with entity classes are more suitable as anchor points for contrastive learning.

## 3.6 Impact of Type-Aware Prototypes

In order to investigate the effectiveness of our proposed strategy for solving the problem of inaccurate and unstable prototypes in the type classification stage, we further perform an analysis of the impact of stability and quality of prototypes. We select three baselines as our compared methods: 1) TadNER w/o Type Names (TN) (the second variant baseline in the ablation study). 2) DecomposedMetaNER (Ma et al., 2022c). 3) Vanilla Contrastive Learning (CL), which adopts token-token contrastive loss and was proposed by Das et al. (2022). We use it to train the type classification

module in a decomposed NER framework, in order to explore whether it can address the issue of unstable and inaccurate prototypes. Here we adopt the same 10 samplings used in the 1-shot Domain Transfer experiments.

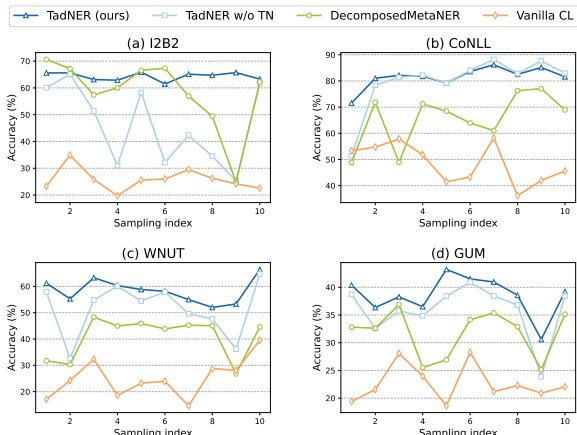

Figure 5: Impacts of prototypes by different methods under 1-shot Domain Transfer setting. The horizontal and vertical coordinates indicate the n-th sampling and the accuracy of type classification, respectively.

As shown in Figure 5, our proposed TadNER achieves a significant improvement over DecomposedMetaNER on each dataset and is more stable across different samplings. Besides, removing type names causes a sharp performance drop in some cases for TadNER w/o TN, indicating that the incorporation of type names indeed helps construct more stable and accurate prototypes. Moreover, Vanilla CL performs extremely poorly due to the introduction of an additional projection layer, which is a crucial component employed in various contrastive learning methods (Chen et al., 2020; Das et al., 2022). However, the inclusion of this layer hampers the model's capacity to acquire adequate semantics related to entity classification.

## 3.7 Error Analysis

We conduct an error analysis to examine the detailed types of errors made by different models. The error statistics are shown in Table 4.

We can observe that: 1) Our TadNER makes fewer errors than baselines overall. Notably, it significantly reduces false negatives, indicating its ability to accurately recall more correct entities. 2) Both TadNER and FSLS can effectively reduce "Type" errors by incorporating type names. However, though FSLS has less "Type" errors than our TadNER, it produces a much larger number of un-

| Models | False Positive | | | | False Negative | | | | False |
|---|---|---|---|---|---|---|---|---|---|
| | Span | | Type | | Span | | Type | | |
| | Num. | Ratio | Num. | Ratio | Num. | Ratio | Num. | Ratio | |
| FSLS | 990 | 85.4% | 169 | 14.6% | 1178 | 87.5% | 169 | 12.5% | 2506 |
| CONTaiNER | 881 | 63.3% | 511 | 36.7% | 628 | 55.1% | 511 | 44.9% | 2531 |
| ESD | 562 | 56.5% | 433 | 43.5% | 932 | 68.3% | 433 | 31.7% | 2360 |
| DecomposedMetaNER | 622 | 56.2% | 485 | 43.8% | 639 | 56.9% | 485 | 43.1% | 2231 |
| **TadNER** | 786 | 81.2% | 182 | 18.8% | 450 | 71.2% | 182 | 28.8% | 1600 |

Table 4: Error analysis for different methods under the Few-NERD Intra 5-way 1∼2-shot setting. We select the first 300 episodes for analysis. "False Positive" and "False Negative" denote the incorrectly extracted entities and unrecalled entities, respectively. "Span" and "Type" denote the error is due to incorrect span/type.

recalled samples, i.e., false negatives. 3) Our Tad-NER still suffers from inaccurate span prediction, which inspires our future work.

### 3.8 Model Efficiency

Compared to one-stage approaches, e.g., CON-TaiNER, two-stage models require more parameters, longer training and inference times. To have a close look at the time cost induced by two-stage models, we perform a model efficiency analysis and show the results in Table 5.

| Paradigms | Models | #Para. | Train | Adapt | Inference | F1 |
|---|---|---|---|---|---|---|
| One-stage | FSLS | 222M | 1871s | 10s | 14ms | 30.38 |
| | CONTaiNER | 112M | 980s | 1s | 17ms | 41.51 |
| Two-stage | ESD | 112M | 2601s | 0s | 35ms | 36.08 |
| | DecomposedMetaNER | 222M | 35495s | 2s | 37ms | 49.48 |
| | **TadNER** | 222M | 3796s | 1.5s | 32ms | 60.29 |

Table 5: Model efficiency analysis for different methods under the Few-NERD Intra 5-way 1∼2-shot setting.

From Table 5, it can be seen that two-stage models indeed require longer training and inference time than one-stage models. However, two-stage models often get better performance. In particular, our TadNER is the most effective one among both one-stage and two-stage models, and it achieves a F1 improvement of 45% and 67% over CON-TaiNER and ESD. It is also the most efficient one among three two-stage models in terms of the inference time.

### 3.9 Zero-Shot Performance

Since there is no domain-specific support set under zero-shot NER settings, it is extremely challenging and rarely explored. While we believe our proposed TadNER can obtain certain zero-shot ability after training in the source domain for the following two reasons: 1) the model can extract entity spans in the span detection stage before fine-tuning with

support samples, 2) since the feature space learnt in the type classification stage is well generalized and type-aware, we can directly adopt the representations of type names as prototypes of novel entity types. To demonstrate the promising performance of our model under zero-shot settings, we select SpanNER (Wang et al., 2021) as a strong baseline, which is a decomposed-based method and good at solving zero-shot NER problem.

| Model | Domain Transfer | | | | |
|---|---|---|---|---|---|
| | I2B2 | CoNLL | WNUT | GUM | Avg. |
| SpanNER (0-shot) | 8.02 | 23.63 | 24.82 | 6.57 | 15.76 |
| **TadNER (0-shot)** | **17.13** | **43.14** | **25.06** | **7.62** | **23.24** |

Table 6: F1 scores under Domain Transfer zero-shot settings.

As shown in Table 6, our proposed TadNER performs better than SpanNER (Wang et al., 2021) under every case. The reason for this may be that the type classification of SpanNER is based on a traditional supervised classification model, which performs worse generalization in cross-domain scenarios. Besides, compared with previous metric-based methods (Das et al., 2022; Ma et al., 2022c) for few-shot NER, which heavily rely on support sets and had **no** zero-shot capability, our method is more inspirational for future zero-shot NER works.

## 4   Related Work

**Few-Shot NER**   Few-shot NER methods can be categorized into two types: prompt-based and metric-based. Prompt-based methods focus on leveraging pre-trained language model knowledge for NER through prompt learning (Cui et al., 2021; Ma et al., 2022b; Huang et al., 2022; Lee et al., 2022). They rely on templates, prompts, or good examples to utilize the pre-trained knowledge effectively. Metric-based methods aim to learn a feature space with good generalizability and classify test samples using nearest class prototypes (Snell et al., 2017; Fritzler et al., 2019; Ji et al., 2022; Ma et al., 2022c) or neighbor samples (Yang and Katiyar, 2020; Das et al., 2022).

There are also some efforts to improve few-shot NER by incorporating type name (label) semantics (Hou et al., 2020; Ma et al., 2022a). These methods usually treat labels as class representatives and align tokens with them, yet neglecting the joint training of entity words and label representations. Hence they can only use either support sets

or labels as class references. Instead, our method exploits support samples and type names simultaneously, which helps construct more accurate and stable prototypes in the target domain.

**Task Decomposition and Contrastive Learning**   Recently, decomposed-based methods have emerged as effective solutions for the NER problem (Shen et al., 2021; Wang et al., 2021; Zhang et al., 2022; Wang et al., 2022; Ma et al., 2022c). These methods can learn entity boundary information well in data-limited scenarios and often get better results. However, the widely used prototypical networks in these methods may encounter inaccurate and unstable prototypes given limited support samples at the type classification stage. Besides, they may face the problem of over-detected false spans produced at the span detection stage. Our method can address these two issues via the proposed type-aware contrastive learning and type-aware span filtering strategies.

Our method is also inspired by contrastive learning (Chen et al., 2020; Khosla et al., 2020). Due to its good generalization performance, two recent methods (Das et al., 2022; Huang et al., 2022) borrow this idea for few-shot NER, which construct contrastive loss between tokens or between the token and the prompt. However, they are both the end-to-end approach and thus have the inherent drawback that cannot learn good entity boundary information. In contrast, our method is a decomposed one and our contrastive loss is constructed between tokens with additional type name information, which can find accurate boundary and learn a type-aware feature space.

## 5   Conclusion

In this paper, we propose a novel TadNER framework for few-shot NER, which handles the span detection and type classification sub-tasks at two stages. For type classification, we present a type-aware contrastive learning strategy to learn a type-aware and generalized feature space, enabling the model to construct more accurate and stable prototypes with the help of type names. Based on it, we introduce a type-aware span filtering strategy for removing over-detected false spans produced at the span detection stage. Extensive experiments demonstrate that our method achieves superior performance over previous SOTA methods, especially in the challenging scenarios. In the future, we will try to extend TadNER to other NLP tasks.

## Limitations

Our proposed TadNER mainly focuses on the type classification stage of few-shot NER and simply adopt token classification for detecting entity spans. There might be better solutions, e.g., using global boundary matrix. However, due to its high GPU memory requirements, we do not include it in our current framework. This drives us to find more efficient and powerful span detector for better few-shot NER performance in the future.

## Ethics Statement

Our work is entirely at the methodological level and therefore there will not be any negative social impacts. In addition, since the performance of the model is not yet at a practical level, it cannot be applied in certain high-risk scenarios (such as the I2B2 dataset used in our paper) yet, leaving room for further improvements in the future.

## Acknowledgments

This work was supported by the grant from the National Natural Science Foundation of China (NSFC) project (No. 62276193). It was also supported by the Joint Laboratory on Credit Science and Technology of CSCI-Wuhan University.

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

**Algorithm 1** Procedure of target domain inference in TadNER.

**Require:** Support set $S_{target}$; Query set $Q_{target}$; Class set $\mathcal{T}_{target}$; Encoders $f_{\theta_1}$, $f_{\theta_2}$;
**Output:** Query set predictions $S_{result}$
1: $\mathcal{L}_{prev} = \infty$; $\mathcal{L}_{prev} \in \mathbb{R}_+$ (Any large positive value);
2: $\mathcal{L}_{span} = \mathcal{L}_{prev} - 1$;
3: **while** $\mathcal{L}_{span} < \mathcal{L}_{prev}$ **do**
4:     $\mathcal{L}_{prev} = \mathcal{L}_{span}$;
5:     Compute loss $\mathcal{L}_{span}$ using Eq. (3);
6:     Update $f_{\theta_2} \rightarrow f_{\theta_2'}$ to reduce $\mathcal{L}_{span}$;
7: **end while**
8: $\mathcal{L}_{prev} = \infty$; $\mathcal{L}_{prev} \in \mathbb{R}_+$ (Any large positive value);
9: $\mathcal{L}_{label} = \mathcal{L}_{prev} - 1$;
10: **while** $\mathcal{L}_{label} < \mathcal{L}_{prev}$ **do**
11:     $\mathcal{L}_{prev} = \mathcal{L}_{label}$;
12:     Compute loss $\mathcal{L}_{label}$ using Eq. (9);
13:     Update parameters $\theta_2 \rightarrow \theta_2'$ to reduce $\mathcal{L}_{label}$;
14: **end while**
15: $C_{span} = \{\}$;
16: **for** $X_i$ in $Q_{target}$ **do**
17:     Extract candidate entity spans $C_{span}^i$ from sentence $X_i$ according to Section 2.2.1;
18:     $C_{span} = C_{span} \cup C_{span}^i$;
19: **end for**
20: Calculate threshold $\gamma_t$ for span filtering using Eq. (11);
21: Calculate all prototypes in $\mathcal{T}_{target}$ using Eq. (12);
22: The prototype of class $t_j$ is denoted as $\mathbf{p_j}$;
    $S_{result} = \{\}$;
23: **for** $s_i$ in $C_{span}$ **do**
24:     $max\_sim = \max\limits_{t_j \in \mathcal{T}_{target}} ((f_{\theta_2'}(s_i) \oplus f_{\theta_2'}(s_i)) \cdot \mathbf{p_j}^\top)$
25:     **if** $max\_sim/2 > \gamma_t$ **then**
26:         Assign the label $y_{pred}$ to $s_i$ using Eq. (14);
27:         $S_{result} = S_{result} \cup \{s_i\}$;
28:     **else**
29:         Remove this candidate span $s_i$;
30:     **end if**
31: **end for**

# A  Appendix

## A.1  Target Domain Inference Algorithm

Algorithm 1 describes the process of domain adaptation using support set in the target domain and prediction on the query set. Lines 1-7 describe the target domain adaptation process for the span detection stage. Lines 8-14 describe the target domain adaptation process for the type classification stage. Lines 15-19 describe the extraction of candidate entity spans in the query set using the fine-tuned span detector. Lines 20-31 describe the candidate entity span filtering and entity type classification using type-aware prototypes.

## A.2  Statistics of Datasets

Table 7 shows statistics of various datasets used in our experiments.

| Dataset | Domain | # Classes | # Sentences | # Entities |
|---------|--------|-----------|-------------|------------|
| Few-NERD | Wikipedia | 66 | 188.2k | 491.7k |
| I2B2'14 | Medical | 23 | 140.8k | 29.2k |
| CoNLL'03 | News | 4 | 20.7k | 35.1k |
| GUM | Wiki | 11 | 3.5k | 6.1k |
| WNUT'17 | Social | 6 | 5.7k | 3.9k |
| OntoNotes | General | 18 | 76.7k | 104.2k |

Table 7: Dataset statistics

## A.3  Details of Evaluation Methods

**Episode-level Evaluation**   Following Ma et al. (2022c), we adopt the episode-level evaluation method for the **Few-NERD** dataset. Each episode consists of a support set and a query set, both given in the n-way k-shot form. In each episode, the model trained in the source domain is tested on the query set by utilizing the support set. To make fair comparisons, we obtain the Micro F1 score with the episode-data processed by Ding et al. (2021). We report the mean F1 score with standard deviation using 3 different seeds.

**Dataset-level Evaluation**   Yang and Katiyar (2020) point that sampling test episodes may not reflect the real-world performance due to various data distributions, and they propose to sample support sets and then test the model in the original test set. Each support set consists of $k$ examples corresponding to each label. The final Micro F1 scores and standard deviations are obtained using different sampled support sets. Thus, following Yang and Katiyar (2020) and Das et al. (2022), we also adopt this evaluation schema for **Domain Transfer** settings. For fair comparisons, we use the support sets sampled by Das et al. (2022) [8].

## A.4  Baselines

**ProtoBERT** (Fritzler et al., 2019) adopts a token-level prototypical network, where the prototype of each class is obtained by averaging token samples of the same label, and the label of each unlabeled token in the query set is determined by its nearest class prototype.
**NNShot** (Yang and Katiyar, 2020) pre-trains BERT by traditional classification methods in the source domain training phase, and decides the class of each unlabeled token by the nearest neighbor at the token level in the target domain inference phase.
**StructShot** (Yang and Katiyar, 2020) is based on NNshot and uses an abstract transition probability for Viterbi decoding during testing.

---

[8] https://github.com/psunlpgroup/CONTaiNER.

**ESD** (Wang et al., 2022) uses a span-level prototypical network, which designs multiple prototypes for O-tokens and uses inter- and cross-span attention for better span representation. **FSLS** (Ma et al., 2022a) adopts two encoders, one for obtaining type names representations and the other for token representations. During the training procedure, the Euclidean distance between tokens and their corresponding type name semantics are minimized. During prediction, the label for a token is determined based on the closest type name semantics. We chose this baseline to demonstrate the superiority of our approach over existing approaches using the semantics of type names. **CONTaiNER** (Das et al., 2022) first trains BERT in the source domain using token-level contrastive learning loss function, then fine-tunes the trained model on the support set, and finally use the nearest neighbor method proposed in NNShot (Yang and Katiyar, 2020) for target domain inference phase. **DecomposedMetaNER** (Ma et al., 2022c) is a decomposed approach that incorporates model-agnostic meta-learning strategy into traditional prototypical network to learn a model-agnostic model and more fully exploits the support set.

## A.5 Implementation Details

Following previous methods (Ding et al., 2021; Das et al., 2022; Ma et al., 2022c), we use `bert-base-uncased` model (Devlin et al., 2019) from HuggingFace (Wolf et al., 2020)[9] as our encoder $f_{\theta_1}$ and $f_{\theta_2}$.

During the source domain training procedure, we use AdamW (Loshchilov and Hutter, 2019) as the optimizer with a learning rate of 3e-5 and 1% linear warmup steps, and the batch size is set to 64. We set the temperature hyper-parameter $\tau = 0.05$ in Eq.(6) and keep dropout rate as 0.2 in the classification layer of the span detection.

As for the early stopping strategy in 2.2.1, we found that the fewer samples face a higher risk of over-fitting, and a lower $\beta$ threshold is required. So we set $\beta = 2$ in all 1-shot settings and $\beta = 6$ in all other cases. Table 8 shows the searching space of each hyper-parameter. Besides, we implement our framework with Pytorch 1.12[10] and train it with a V100-16G GPU.

---

[9]https://huggingface.co/bert-base-uncased
[10]https://pytorch.org/

| Learning rate | {1e-5, 3e-5, 1e-4} |
|---|---|
| Batch size | { 32, 64, 128} |
| Dropout rate | {0.1, 0.2, 0.5} |
| temperature $\tau$ | {0.01, 0.05, 0.1} |
| Early stopping threshold $\beta$ | {1, 2, 4, 6, 8} |

Table 8: Hyper-parameters search space in our experiments.

## A.6 Analysis of Tagging Schemes in the Span Detection Stage

Table 9 and Table 10 show the span detection and overall performance under the Domain Transfer settings. We observe that: 1) The three tagging schemes have their own advantages and disadvantages. IO and BIO schemes can achieve higher recall, BIOES can achieve higher precision. 2) The IO tagging scheme can achieve the best overall performance in most settings, except for the GUM dataset. Therefore, the IO scheme is selected for the span detection stage in this paper. 3) The type-aware span filtering strategy proposed in this paper shown robust and positive effects across different tagging schemes. Even when dealing with entity-dense datasets, where incorrect entity spans are minimal, this strategy does not significantly impair performance. In future work, we can try to combine the advantages and disadvantages of different tagging schemes to further improve the performance of the span detection stage.

## A.7 Detailed Type Names

Tables 11 and 12 show the type names used in our TadNER framework. Tables 13 and 14 show the variant type names used in the analysis experiments on the impact of type names in Section 3.5.

| Stage | Filtered | Schema | I2B2 | | | | | | CoNLL | | | | | |
|---|---|---|---|---|---|---|---|---|---|---|---|---|---|---|
| | | | 1-shot | | | 5-shot | | | 1-shot | | | 5-shot | | |
| | | | Precision | Recall | F1 | Precision | Recall | F1 | Precision | Recall | F1 | Precision | Recall | F1 |
| Span | No | IO | 19.62 | 70.59 | 30.46 | 25.12 | 77.71 | 37.86 | 75.05 | 84.28 | 78.96 | 87.48 | 90.68 | 89.01 |
| | | BIO | 19.84 | 67.89 | 30.49 | 22.36 | 75.76 | 34.40 | 72.01 | 84.27 | 77.15 | 85.87 | 88.78 | 87.24 |
| | | BIOES | 19.71 | 60.46 | 29.47 | 23.89 | 70.19 | 35.53 | 70.38 | 80.93 | 74.89 | 84.02 | 87.77 | 85.72 |
| | Yes | IO | 53.79 | 41.54 | 45.33 | 55.78 | 52.84 | 53.82 | 78.65 | 83.25 | 80.47 | 87.86 | 89.56 | 88.67 |
| | | BIO | 54.20 | 40.83 | 45.63 | 53.24 | 55.64 | 53.63 | 77.22 | 84.29 | 80.18 | 87.11 | 88.78 | 87.90 |
| | | BIOES | 52.77 | 34.04 | 39.80 | 57.46 | 50.97 | 53.32 | 74.39 | 80.72 | 77.00 | 84.65 | 87.65 | 86.00 |
| Span+Type | No | IO | 14.14 | 47.18 | 21.57 | 17.83 | 51.11 | 26.35 | 65.37 | 72.73 | 68.47 | 79.06 | 81.32 | 80.14 |
| | | BIO | 14.92 | 49.32 | 22.74 | 16.65 | 54.40 | 25.40 | 63.66 | 74.08 | 68.01 | 77.88 | 80.27 | 79.00 |
| | | BIOES | 14.18 | 42.01 | 21.02 | 17.44 | 49.36 | 25.69 | 61.84 | 70.69 | 65.62 | 76.36 | 79.46 | 77.75 |
| | Yes | IO | 47.24 | 35.77 | 39.32 | 46.92 | 44.33 | 45.20 | 68.89 | 72.70 | 70.38 | 79.81 | 81.31 | 80.53 |
| | | BIO | 47.83 | 35.42 | 39.87 | 45.18 | 47.00 | 45.39 | 67.98 | 74.07 | 70.52 | 78.75 | 80.26 | 79.47 |
| | | BIOES | 45.47 | 28.69 | 33.80 | 47.54 | 42.15 | 44.10 | 65.26 | 70.62 | 67.46 | 76.80 | 79.46 | 77.99 |

Table 9: span detection.

| Stage | Filtered | Schema | WNUT | | | | | | GUM | | | | | |
|---|---|---|---|---|---|---|---|---|---|---|---|---|---|---|
| | | | 1-shot | | | 5-shot | | | 1-shot | | | 5-shot | | |
| | | | Precision | Recall | F1 | Precision | Recall | F1 | Precision | Recall | F1 | Precision | Recall | F1 |
| Span | No | IO | 38.42 | 65.42 | 47.37 | 40.70 | 65.64 | 49.13 | 45.93 | 45.70 | 45.72 | 56.41 | 64.25 | 60.04 |
| | | BIO | 40.89 | 63.85 | 48.82 | 38.28 | 68.60 | 48.92 | 44.86 | 45.67 | 45.05 | 53.99 | 64.34 | 58.64 |
| | | BIOES | 42.67 | 56.14 | 47.30 | 41.90 | 65.24 | 50.59 | 54.28 | 48.32 | 50.97 | 60.57 | 64.07 | 62.22 |
| | Yes | IO | 40.86 | 63.50 | 48.49 | 41.13 | 65.27 | 49.34 | 46.14 | 44.61 | 45.26 | 55.98 | 62.09 | 58.84 |
| | | BIO | 43.61 | 61.33 | 49.41 | 38.74 | 68.15 | 49.17 | 45.55 | 45.74 | 45.44 | 54.74 | 64.40 | 59.11 |
| | | BIOES | 45.78 | 54.39 | 48.14 | 42.45 | 65.06 | 50.92 | 54.58 | 47.92 | 50.88 | 60.88 | 63.56 | 62.15 |
| Span+Type | No | IO | 25.86 | 43.19 | 31.62 | 28.59 | 45.35 | 34.26 | 24.64 | 23.90 | 24.21 | 33.39 | 37.02 | 35.09 |
| | | BIO | 27.34 | 42.11 | 32.52 | 25.69 | 45.83 | 32.77 | 24.97 | 25.24 | 24.99 | 33.14 | 39.04 | 35.81 |
| | | BIOES | 28.94 | 36.92 | 31.74 | 28.60 | 44.37 | 34.48 | 30.20 | 26.65 | 28.23 | 37.38 | 39.02 | 38.15 |
| | Yes | IO | 27.95 | 42.60 | 32.84 | 28.95 | 45.34 | 34.51 | 24.65 | 23.87 | 24.20 | 33.39 | 37.02 | 35.09 |
| | | BIO | 29.79 | 41.22 | 33.56 | 26.06 | 45.80 | 33.06 | 24.98 | 25.23 | 24.99 | 33.14 | 39.04 | 35.81 |
| | | BIOES | 31.72 | 36.31 | 32.85 | 28.95 | 44.37 | 34.72 | 30.21 | 26.64 | 28.22 | 37.38 | 39.02 | 38.15 |

Table 10: span detection.

| Dataset | Labels | Type names |
|---|---|---|
| | art-broadcastprogram | broadcast program |
| | art-film | film |
| | art-music | music |
| | art-other | other art |
| | art-painting | painting |
| | art-writtenart | written art |
| | person-actor | actor |
| | person-artist/author | artist author |
| | person-athlete | athlete |
| | person-director | director |
| | person-other | other person |
| | person-politician | politician |
| | person-scholar | scholar |
| | person-soldier | soldier |
| | product-airplane | airplane |
| | product-car | car |
| | product-food | food |
| | product-game | game |
| | product-other | other product |
| | product-ship | ship |
| | product-software | software |
| | product-train | train |
| | product-weapon | weapon |
| | other-astronomything | astronomy thing |
| | other-award | award |
| | other-biologything | biology thing |
| | other-chemicalthing | chemical thing |
| | other-currency | currency |
| | other-disease | disease |
| | other-educationaldegree | educational degree |
| | other-god | god |
| | other-language | language |
| | other-law | law |
| | other-livingthing | living thing |
| | other-medical | medical |
| | building-airport | airport |
| | building-hospital | hospital |
| | building-hotel | hotel |
| | building-library | library |
| | building-other | other building |
| | building-restaurant | restaurant |
| | building-sportsfacility | sports facility |
| | building-theater | theater |
| | event-attack/battle /war/militaryconflict | attack battle war military conflict |
| | event-disaster | disaster |
| | event-election | election |
| | event-other | other event |
| | event-protest | protest |
| Few-NERD | event-sportsevent | sports event |
| | location-bodiesofwater | bodies of water |
| | location-GPE | geographical social political entity |
| | location-island | island |
| | location-mountain | mountain |
| | location-other | other location |
| | location-park | park |
| | location-road/railway /highway/transit | road railway highway transit |
| | organization-company | company |
| | organization-education | education |
| | organization-government /governmentagency | government agency |
| | organization-media/newspaper | media newspaper |
| | organization-other | other organization |
| | organization-politicalparty | political party |
| | organization-religion | religion |
| | organization-showorganization | show organization |
| | organization-sportsleague | sports league |
| | organization-sportsteam | sports team |

Table 11: Original labels and their corresponding natural-language-form type names of Few-NERD.

| Dataset | Labels | Type names |
|---|---|---|
| | AGE | age |
| | BIOID | biometric ID |
| | CITY | city |
| | COUNTRY | country |
| | DATE | date |
| | DEVICE | device |
| | DOCTOR | doctor |
| | EMAIL | email |
| | FAX | fax |
| | HEALTHPLAN | health plan number |
| | HOSPITAL | hospital |
| I2B2'14 | IDNUM | ID number |
| | LOCATION_OTHER | location |
| | MEDICALRECORD | medical record |
| | ORGANIZATION | organization |
| | PATIENT | patient |
| | PHONE | phone number |
| | PROFESSION | profession |
| | STATE | state |
| | STREET | street |
| | URL | url |
| | USERNAME | username |
| | ZIP | zip code |
| | PER | person |
| CoNLL'03 | LOC | location |
| | ORG | organization |
| | MISC | miscellaneous |
| | abstract | abstract |
| | animal | animal |
| | event | event |
| | object | object |
| | organization | organization |
| GUM | person | person |
| | place | place |
| | plant | plant |
| | quantity | quantity |
| | substance | substance |
| | time | time |
| | corporation | corporation |
| | creative-work | creative work |
| WNUT'17 | group | group |
| | location | location |
| | person | person |
| | product | product |
| | CARDINAL | cardinal |
| | DATE | date |
| | EVENT | event |
| | FAC | fac |
| | GPE | geographical social political entity |
| | LANGUAGE | language |
| | LAW | law |
| | LOC | location |
| | MONEY | money |
| Ontonotes | NORP | nationality religion |
| | ORDINAL | ordinal |
| | ORG | organization |
| | PERCENT | percent |
| | PERSON | person |
| | PRODUCT | product |
| | QUANTITY | quantity |
| | TIME | time |
| | WORK_OF_ART | work of art |

Table 12: Original labels and their corresponding natural-language-form type names of datasets under Domain Transfer settings.

| Original Type Names | Synonym 1 | Synonym 2 | Synonym 3 |
|---|---|---|---|
| broadcast program | television show | TV program | telecast |
| film | movie | motion picture | cinema |
| music | melody | tunes | songs |
| other art | different art | alternative art | diverse art |
| painting | artwork | canvas | picture |
| written art | literature | written work | prose |
| actor | performer | thespian | artist |
| artist author | creative writer | author | wordsmith |
| athlete | sportsman/woman | player | competitor |
| director | filmmaker | supervisor | manager |
| other person | someone else | another person | another individual |
| politician | statesman/woman | lawmaker | public servant |
| scholar | academic | intellectual | researcher |
| soldier | military personnel | serviceman/woman | trooper |
| airplane | aircraft | plane | jet |
| car | automobile | nourishment | fare |
| food | cuisine | nourishment | fare |
| game | sport | competition | match |
| other product | different product | alternative item | various commodity |
| ship | vessel | boat | craft |
| software | program | application | computer program |
| train | locomotive | railway vehicle | railcar |
| weapon | armament | firearm | arm |
| astronomy thing | celestial object | astronomical entity | heavenly body |
| award | accolade | prize | recognition |
| biology-thing | biological entity | living organism | life form |
| chemical thing | chemical substance | compound | element |
| currency | money | cash | legal tender |
| disease | illness | sickness | disorder |
| educational degree | academic qualification | diploma | certification |
| god | deity | divine being | higher power |
| language | tongue | speech | communication |
| law | legislation | legal system | jurisprudence |
| living thing | organism | creature | being |
| medical | healthcare | medicinal | therapeutic |
| bodies of water | Waterways | aquatic features | lakes and rivers |
| geographical social political entity | Territory | region | jurisdiction |
| island | Isle | islet | key |
| mountain | Peak | summit | range |
| other location | Site | spot | place |
| park | Garden | reserve | recreational area |
| road railway highway transit | Route | thoroughfare | transportation network |
| company | Corporation | firm | enterprise |
| education | Learning | schooling | instruction |
| government agency | Public body | administrative department | authority |
| media newspaper | Press | journalism | news organization |
| other organization | Institution | establishment | association |
| political party | Political group | faction | party organization |
| religion | Faith | belief system | spirituality |
| show organization | Production company | entertainment group | performance troupe |
| sports league | Athletic association | sporting federation | league organization |
| sports team | Athletic club | competitive squad | sporting roster |

Table 13: Variant type names for Few-NERD Intra setting.

| Original Type Names | Synonym 1 | Synonym 2 | Synonym 3 |
|---|---|---|---|
| cardinal | Primary | fundamental | principal |
| date | Day | time | appointment |
| event | Occasion | happening | function |
| fac | Facility | building | structure |
| geographical social political entity | Territory | region | jurisdiction |
| language | Tongue | speech | communication |
| law | Regulation | rule | statute |
| location | Place | site | spot |
| money | Currency | funds | finances |
| nationality religion | Citizenship | faith | belief system |
| ordinal | Sequential | numbered | ordered |
| organization | Institution | establishment | association |
| percent | Percentage | proportion | rate |
| person | Individual | human | character |
| product | Item | good | merchandise |
| quantity | Amount | volume | measure |
| time | Duration | period | interval |
| work of art | Artwork | creation | masterpiece |
| person | Individual | human being | somebody |
| location | Place | site | spot |
| organization | Institution | establishment | company |
| miscellaneous | Various | diverse | mixed |

Table 14: Variant type names for Domain Transfer setting. Here we show the type names of the OntoNotes dataset and the CoNLL2003 dataset.