# OpenReview forum: "Type-Aware Decomposed Framework for Few-Shot Named Entity Recognition"
_EMNLP/2023/Conference — EMNLP 2023 Findings_

### Official Review · Reviewer_NUFY · 2023-07-31

**Soundness:** 4

**Excitement:**

4: Strong: This paper deepens the understanding of some phenomenon or lowers the barriers to an existing research direction.

**Paper Topic And Main Contributions:**

The authors propose a method for few-shot NER build on top of two-stage methods where first span detection is performed followed by span classification. However, they improve over recent methods by incorporating: (1) an effective way of exploiting the text representations of the labels by embedding them into a common space with the entity tokens, and (2) using these embeddings to filter out false entity span candidates that are too dissimilar to the provided labels. Primarily, they construct entity prototype representations by embedding the entity tokens and the mapped class labels with the same encoder and concatenate them in two variants (i.e. in both directions: entity-label and label-entity). Then, the encoder is trained with a contrastive learning objective that pulls these representations together if they have the same label and pushes them away from each other, if this is not the case.

The authors show that this approach is very effective by conducting few-shot experiments on several datasets (Few-NERD and others) where they outperform previous work (one-stage and two-stage) on a large margin, especially in the realm of very few (1 ~ 2) training examples. Furthermore, they include an extensive ablation study (effect of: span filtering, incorporating the label text, fine-tuning of the encoders for each stage on the support set) and a quantitative as well as a qualitative error analysis.

**Questions For The Authors:**

 - Q1: In Table 2, where did you get the scores marked as "from Ma et. al (2022c)" from? Maybe I'm mistaken, but I can not find these values in the publication: https://aclanthology.org/2022.findings-acl.124/
 - Q2: How is fθ2 defined? Is it really the same as fθ1? But it looks like Map(yi) = t′i can be a token sequence (some of the mapped types in the appendix consist of multiple words). But in this case, equation (4) seems to only work, if fθ1 maps to R^r... in section 2.1.2, it is said that individual entity tokens ei are embedded with fθ2. However, in "2.2.2 Type-Aware Span Filtering" it is said that spans are filtered. Does the filtering happen token-wise? Or are all tokens of one span embedded at once and the result is then filtered (same  for the "Inference" section, in both sections you talk about spans)? But if fθ2 takes really a token sequence as input, Figure 2 is very misleading because "Barack Obama" is split into two training instances (("Barack", "person") o ("person", "Barack")) and (("Obama", "person" o ("person", "Obama")) where I had just assumed (("Barack Obama", "person") o ("person", "Barack Obama")).
 - Q3: Since your method relies on useful label names: What was the effort to create the respective label name mappings? Could you give some points that this is negligible?

**Reasons To Accept:**

The authors propose an effective method for few-shot NER that significantly outperforms previous approaches, especially for the very few shot setting. They explain their method quite well and conduct reasonable experiments to show the effectiveness of their method.

**Reasons To Reject:**

 - see Q1 (re-reported scores seem to be lower than in the cited paper)
 - see Q2 (definition of fθ2 is unclear)

**Reproducibility:**

5: Could easily reproduce the results.

**Reviewer Confidence:**

4: Quite sure. I tried to check the important points carefully. It's unlikely, though conceivable, that I missed something that should affect my ratings.

**Typos Grammar Style And Presentation Improvements:**

 - typo: line 127 "datset"
 - type: line 938 "doamin"

---

> ### Author Rebuttal · Authors · 2023-08-28
>
> We thank the reviewer for the constructive comments. We hope the following clarifications can address the reviewer’s concerns.
> > Q1. In Table 2, where did you get the scores marked as "from Ma et. al (2022c)" from? Maybe I'm mistaken, but I can not find these values in the publication: https://aclanthology.org/2022.findings-acl.124/
>
> This problem is caused by the different versions of the Few-NERD dataset [1] (corresponding to arXiv-V5 and arXiv-V6, respectively).
> On the publicly available leaderboard (link: https://ningding97.github.io/fewnerd/) by the Few-NERD authors, the Few-NERD arXiv-V5 data was deleted by the authors and was replaced with the Few-NERD arXiv-V6 version when we conducted our experiements, so we had to use the Few-NERD arXiv-V6 version.
>
> Results in the camera ready version of Ma et. al's (2022c) [2] paper were obtained using the Few-NERD arXiv-V5 data.
> For the convenience of comparison with their results, Ma et. al (2022c) [2] reproduced the results using the Few-NERD arXiv-V6 data and published the results on Github. The link is: https://github.com/microsoft/vert-papers/tree/master/papers/DecomposedMetaNER#few-nerd-arxiv-v6-version, which is also the source of results in our paper which are marked as "from Ma et. al (2022c)".
>
> Due to the space limit, we have to place this explanation in footnote 7 of Appendix A.3.
> To avoid potential confusion for the readers, we will include this explanation to the main body of the paper.
>
> [1] Ding et al., 2021. Few-NERD: A Few-shot Named Entity Recognition Dataset. ACL-IJCNLP 2021.\
> [2] Ma et al., 2022. Decomposed Meta-Learning for Few-Shot Named Entity Recognition. ACL 2022, pages 1584–1596.
>
>
> > Q2. How is $f_{θ_2}$ defined? Is it really the same as $f_{θ_1}$? \
> But it looks like Map($y_i$) = $t_{i}^{'}$ can be a token sequence (some of the mapped types in the appendix consist of multiple words). \
> But in this case, equation (4) seems to only work, if $f_{θ_1}$ maps to $R^r$... in section 2.1.2, it is said that individual entity tokens ei are embedded with $f_{θ_2}$. \
> However, in "2.2.2 Type-Aware Span Filtering" it is said that spans are filtered. Does the filtering happen token-wise? Or are all tokens of one span embedded at once and the result is then filtered (same for the "Inference" section, in both sections you talk about spans)? \
> But if fθ2 takes really a token sequence as input, Figure 2 is very misleading because "Barack Obama" is split into two training instances (("Barack", "person") o ("person", "Barack")) and ("Obama", "person" o ("person", "Obama")) where I had just assumed (("Barack Obama",
> "person") o ("person", "Barack Obama")).
>
> Sorry for the confusion due to the unclear formalization of $f_{θ_1}()$ and $f_{θ_2}()$ in the paper. \
> Generally speaking, $f_{θ_1}()$ and $f_{θ_2}()$ are the same context encoder with individual parameters.
> Here, we will formalize $f_{θ_1}()$ and $f_{θ_2}()$ in detail and explain the reviewer's questions one by one. For simplicity of description, let's collectively refer to $f_{θ_1}()$ and $f_{θ_2}()$ as $f_{θ}()$.
>
> For the input $X$, the output is $V=f_θ(X)$.
> 1. If $X$ is a token, then $V$ is the context embedding vector of this token.
> 2. If $X$ is a word or span composed of N tokens, i.e., the type name "nationality religion" or entity span "Barack Obama", $V$ is the average of the context embedding vectors for these N tokens.
> 3. If $X$ is a token sequence composed of M tokens $[x_1,…,x_M]$, like the sequence ["Barack", "Obama", "was",…, "1961"] in Figure 2, then V is the sequence of context embeddings $[emb_1,…,emb_M]$=[$f_{θ}$(Barack),…,$f_{θ}$(1961)], where the dimension of $emb_1,…,emb_M$ is the hidden layer dimension of the encoder $f_θ()$.
>
> Based on this formalization, below are our explanations for your questions:
>
> 1\. "Does the filtering happen token-wise? Or are all tokens of one span embedded at once and the result is then filtered."  \
> **Explanation**: The filtering process is span-based. The entire span is inputted to the encoder $f_{θ_2}()$ as a whole (corresponding to the second type of input to $f_{θ}()$ in the formalization above).
>
> 2\. "Figure 2 is very misleading because 'Barack Obama' is split into two training instances"\
> **Explanation**: The input to $f_{θ_2}()$ in Figure 2 is a token sequence, so its output is an embedding sequence where each token $e_i$ corresponds to an output embedding $f_{θ_2}(e_i)$ (corresponding to the third type of input to $f_{θ}()$ in the formalization above).
>
> To avoid any potential confusion, we will add this formalization of the encoders $f_{θ_1}()$ and $f_{θ_2}()$ into the methodology section.
>
> > Q3. Since your method relies on useful label names: What was the effort to create the respective label name mappings? Could you give some points that this is negligible?
>
> The type name mappings for all the datasets used in our experiments can be seen in Tables 11 and 12 of the Appendix A.8. \
> As shown in Table 11, for the Few-NERD dataset, we directly use the fine-grained part of the original label as the corresponding type name, such as "art-film"->"film".
> As shown in Table 12, for the datasets under the Domain Transfer settings, we convert the majority of labels into their most direct natural language form of type names, such as "PER"->"person" and "AGE"->"age".\
> Therefore, the additional effort required to construct the mappings is almost negligible.
>
>
> > Typos: line 127 "datset"; line 938 "doamin"
>
> Thanks a lot for your careful reading, we will fix them per your suggestion.

---

### Official Review · Reviewer_RZMJ · 2023-08-03

**Soundness:** 4

**Excitement:**

3: Ambivalent: It has merits (e.g., it reports state-of-the-art results, the idea is nice), but there are key weaknesses (e.g., it describes incremental work), and it can significantly benefit from another round of revision. However, I won't object to accepting it if my co-reviewers champion it.

**Paper Topic And Main Contributions:**

This paper describes TadNER, a type-aware decomposed framework for named-entity recognition (NER). Its main goal is the enhancement of performance in few-shot NER, i.e. the extension of the set of spans and labels using only few examples.
The authors deal with the problem by two points of view: (i) filter out spans semantically far from the accepted type names; (ii) build stable prototypes of labels, to avoid deviation form the class centers.

**Reasons To Accept:**

The goal of the paper is clear, and the approach seems promising, although it is an application of ideas that already exist for the same kind of task. The use of a big number of baselines and the strong evaluation proposed bring robustness to the whole process.
The most interesting part of the study, in my opinion, is the semantic connection between the transformers and the labels (as described in Appendix A.8).

**Reasons To Reject:**

In some parts, the paper is really hard to follow. In method section, the only real examples are presented in the (small) picture, and it is hard to connect each formula and description to the phases in the image.
In general, the LaTeX formatting should be more tidy (see for example Section 2.1.1).

**Reproducibility:**

4: Could mostly reproduce the results, but there may be some variation because of sample variance or minor variations in their interpretation of the protocol or method.

**Reviewer Confidence:**

3: Pretty sure, but there's a chance I missed something. Although I have a good feel for this area in general, I did not carefully check the paper's details, e.g., the math, experimental design, or novelty.

**Typos Grammar Style And Presentation Improvements:**

There are a lot of errors due to the wrong use of equation in LaTeX (main of them can be found in Section 2.1.1).

---

> ### Author Rebuttal · Authors · 2023-08-28
>
> We thank the reviewer for the constructive comments. We hope the following clarifications can address the reviewer’s concerns.
> > Q1. In some parts, the paper is really hard to follow. In method section, the only real examples are presented in the (small) picture, and it is hard to connect each formula and description to the phases in the image. In general, the LaTeX formatting should be more tidy (see for example Section 2.1.1).
>
> Sorry for the inconvenience, we will enrich the caption in Figure 2 so that the readers can connect the formula with the real example in the picture more clearly.
>
> > Typos: There are a lot of errors due to the wrong use of equation in LaTeX (main of them can be found in Section 2.1.1).
>
> Thanks for your suggestion! We will fix them per your suggestion.

---

### Official Review · Reviewer_WiN1 · 2023-08-04

**Soundness:** 3

**Excitement:**

3: Ambivalent: It has merits (e.g., it reports state-of-the-art results, the idea is nice), but there are key weaknesses (e.g., it describes incremental work), and it can significantly benefit from another round of revision. However, I won't object to accepting it if my co-reviewers champion it.

**Paper Topic And Main Contributions:**

The paper proposes a novel approach called TadNER for few-shot named entity recognition (NER), which aims to overcome two main challenges in this task: over-detected false spans and inaccurate and unstable prototypes. TadNER incorporates a type-aware span filtering strategy and a type-aware contrastive learning strategy.

1)The type-aware span filtering strategy is designed to filter out false spans by identifying and removing those that are semantically far away from type names. This is achieved by considering the distance between each span and the nearest type name and filtering out those that exceed a predefined threshold.

2)The type-aware contrastive learning strategy is designed to construct more accurate and stable prototypes by leveraging type names and support samples as references. This is achieved by jointly optimizing a contrastive loss function that encourages the prototypes to be close to their corresponding support samples and far from other samples in the same type or other types.

3)The proposed TadNER approach is evaluated on various benchmarks, and the results demonstrate that it achieves a new state-of-the-art performance, outperforming existing approaches.

**Questions For The Authors:**

In the reading of this paper, there are the following questions:
1)It seems that on the basis of the mainstream span method, the paper only adds label names to inject information into the model, but does not carry out greater innovation in the method or model. The results of the experiment appear to be improved, but there is no good formula or theoretical basis for the effectiveness of the information added or removed.
2)The article proposes to delete the type far from the type name. In the example Figure 1 given in the article, there is only one word between 1976 and California. If we want to delete the influence of 1976, the window distance is very small, which seems to be helpful only for this particular case, but for most of the samples, Whether it is appropriate, and whether it will cause some important boundary information to be missed. Because in most cases, we want to be able to detect more information from a long text.
3)The paper wants to use type names to obtain richer semantics and prototype representation, but in contrast, the mainstream large models such as GPT contain richer semantics. What is the semantic advantage of the proposed type names compared with the big model? Besides, why does the paper not use the big model to get the semantics?
4)The article's description of the type of deletion is too simple, and there is no detailed description of the basis and distance of deletion, so there is no way to judge the effectiveness of this step. The article seems to give the threshold value in Formula 11 but does not explain the value in the formula, indicating that the description is very vague.
5)Regarding the fine-tuning proposed in the article, may I ask how to divide the support data of the test set during the fine-tuning process? In the case of 1shot, it cannot be divided and fine-tuned the support, which will lead to type loss. How the fine-tuning proposed in the article solves this problem is not explained in the article.
6)The article argues that adding type names can be helpful for prototyping. In the result and analysis part of the article, only the influence on acc after adding the name is given, and the effect of the prototype is not intuitively seen. It seems that the article should give a vector diagram of the prototype to illustrate the correction or aggregation effect of the prototype.
7) Is the model proposed in this paper an experiment based on the DecomposedMetaNER model? If so, is the classification of the training set, verification set, and test set the same as that of DecomposedMetaNER? Can you show more details of the processing

**Reasons To Accept:**

The paper demonstrates a clear structure and presents promising experimental results.

**Reasons To Reject:**

1) The overall logic of the paper is relatively clear, but this paper lacks sufficient novelty and appears to be a limited improvement over the DecomposedMetaNER paper.
2) In the introduction of part of the work, the article lacks details and does not elaborate clearly.
These issues will be elaborated on in the following question and answer section.

**Reproducibility:**

2: Would be hard pressed to reproduce the results. The contribution depends on data that are simply not available outside the author's institution or consortium; not enough details are provided.

**Reviewer Confidence:**

5: Positive that my evaluation is correct. I read the paper very carefully and I am very familiar with related work.

---

> ### Author Rebuttal · Authors · 2023-08-28
>
> We thank the reviewer for the constructive comments. We hope the following clarifications can address the reviewer’s concerns.
> > Q1. It seems that on the basis of the mainstream span method, the paper only adds label names to inject information into the model, but does not carry out greater innovation in the method or model. \
> The results of the experiment appear to be improved, but there is no good formula or theoretical basis for the effectiveness of the information added or removed.
>
> 1\. To the best of our knowledge, no related work has explicitly identified and addressed the issues of "over-detected false spans" and "inaccurate and unstable prototypes" in the context of few-shot NER. In other words, these are new problems for few-shot NER. In order to solve these two issues, we propose to inject type name information and design the type-aware span filtering strategy and the type-aware contrastive learning strategy, respectively.
>
> Although our approach may seem simple and straightforward, this does not necessarily imply a lack of innovation.
> Firstly, recognizing a new problem is always critical to the research community, and it might be more important than solving the problem itself.
> Secondly, our recognized two problems may appear in the few-shot scenarios of other NLP tasks, such as the few-shot event detection task (extracting the event trigger word first and then classifying the event type based on the trigger word).
> Our discoveries and solutions could provide some insights to the researchers when tackling these tasks, too.
>
> 2\. Early probabilistic models such as Semi-CRFs [1] had good formalized analysis. Recent data driven deep learning methods can achieve better performance yet often lacking the theoretical basis. To compensate for this shortcoming, for the added type name information, we conducted as much experimental analysis as possible. As can be seen, in Section 3.5, there are a large number of experiments with "Synonyms", "Meaningless" and "Misleading" variants of type names to demonstrate the effectiveness of type name injection.
>
> [1] Sarawagi S, Cohen W W. Semi-markov conditional random fields for information extraction. Advances in neural information processing systems, 2004, 17.
>
> > Q2. The article proposes to delete the type far from the type name. In the example Figure 1 given in the article, there is only one word between 1976 and California. If we want to delete the influence of 1976, the window distance is very small, which seems to be helpful only for this particular case, but for most of the samples, Whether it is appropriate, and whether it will cause some important boundary information to be missed. Because in most cases, we want to be able to detect more information from a long text.
>
> We guess that the reviewer might have misunderstood the main reason why "1976" was misclassified as "LOC".
> The reviewer may think that the misclassification of "1976" into "LOC" was influenced by the semantic of the nearby token "California". However, the error is mainly due to the bias brought by the domain gap in the few-shot scenario.
>
> First of all, we are focusing on the NER task setting with a fixed set of entity types (which is also the most common in its practical applications). For the example in Figure 1 (a) and (b), we only aim to extract entities of the "ORG" and "LOC" type. So even if "1976" is an entity of the "DATE" type, we don’t want to extract it.
>
> Secondly, assuming under a specific few-shot NER setting (Figure 2), the entity type sets of the training and test sets are {"PER", "DATE"} and {"ORG", "LOC"}, respectively. The model needs to be trained on the training set first, and then be fine-tuned using few support samples in the test set.
> Since entities of the "DATE" type have appeared in the training set, the span detection module will be trained to extract the span of the "DATE" type entity. During the inference on the test set, as shown in the Figure 1 (a), "1976" is extracted at the span detection stage.
> Note that **for this test set, our predefined entity type set does not include the "DATE" type**, i.e., we do not want to retain the entity span of "1976". Continuing to classify it would force it to be assigned an entity type of either "ORG" or "LOC" (as these are the only two types in this test set), resulting in an obviously incorrect classification result. Therefore, the type-aware span filtering strategy is designed to alleviate this problem.
>
> > Q3. The paper wants to use type names to obtain richer semantics and prototype representation, but in contrast, the mainstream large models such as GPT contain richer semantics. \
> What is the semantic advantage of the proposed type names compared with the big model? Besides, why does the paper not use the big model to get the semantics?
>
>
> **Firstly**, we suppose the reviewer's concern is about "using a larger model to get a richer semantics may perform better than incorporating type names". However, the two main issues we've identified and tried to address, i.e., "over-detected false spans" and "inaccurate and unstable prototypes", are not caused by the lack of rich semantic information in the LM.
> Instead, they are caused by the bias from the domain gap and the scarcity of samples under few-shot settings, respectively.
>
> For the issue of "over-detected false spans", the main reason is that at the entity span detection stage, the span information learned by the model from the source domain is inevitably transferred to the target domain. When the entity type sets of the source domain and target domain are different, some entity spans that only belong to the source domain type set rather than the target domain type set will be incorrectly extracted (this is also discussed in question 2 above). Therefore, this problem is due to the bias from the domain gap, not a lack of rich semantic information obtained through the LM.
>
> For the issue of "inaccurate and unstable prototypes", the main reason is that the prototype under the few-shot settings is constructed with very few samples (1/5-shot), which may deviate from the actual category center in some scenarios (Figure 1c).
> This would lead to a reduction in the model's performance (Section 3.6). Hence, even using a larger model to obtain the representation of entity words, there may still exist such a prototype bias caused by the scarcity of samples.
>
> For these two reasons, we think it is unnecessary to conduct experiments with a big model just for obtaining richer semantics.
> Therefore, we only used the BERT model in our experiments, which is frequently adopted by community researchers, including all the baselines we compared, to validate the effectiveness of our proposed method.
>
> **Secondly**, we are not sure whether the reviewer's concern is about the necessity of small PLMs like BERT for this problem. If this is the case, the answer is YES because LLMs cannot perform better than fine-tuned SLMs, e.g., 5-shot + settings in our task, due to the lack of source-domain NER data. A newly published paper [1] in ACL 23 also get the similar conclusion with the help of a large corpus in the source domain.
>
> [1] Chen et al., 2023. Learning In-context Learning for Named Entity Recognition. ACL 2023, pages 13661–13675.
>
>
> > Q4. The article's description of the type of deletion is too simple, and there is no detailed description of the basis and distance of deletion, so there is no way to judge the effectiveness of this step. \
> The article seems to give the threshold value in Formula 11 but does not explain the value in the formula, indicating that the description is very vague.
>
> Sorry for the confusion caused by the lack of detailed descriptions, which is mainly due to space constraints that prevent us from elaborating on the corresponding information in the main body.
>
> 1. For detailed process of the type-aware span filtering (deletion), please refer to lines 20-31 of the algorithm in Appendix A.1. Per your suggestion, we will add a more detailed description of the type-aware span filtering strategy in the corresponding methodology section.
>
> 2. The symbols used in Eq. 11 were mentioned in the previous paragraphs. Specifically, $e_i^s$ refers to the $i_{th}$ entity token in the support set (line 130), and the Map() function is for converting the original class labels into corresponding type names, e.g., "PER"->"person" (page 3, footnote 4). The main purpose of this formula is to obtain a type name-aware threshold using samples in the support set, which would be used for the false span filtering. We will add detailed descriptions of these symbols in Eq. 11 to make it more clear to read.
>
> > Q5. Regarding the fine-tuning proposed in the article, may I ask how to divide the support data of the test set during the fine-tuning process?\
> In the case of 1 shot, it cannot be divided and fine-tuned the support, which will lead to type loss. How the fine-tuning proposed in the article solves this problem is not explained in the article.
>
> We suppose the reviewer's concern about "how to divide the support set into a training set and a validation set" during the fine-tuning process.
>
> In fact, for the few-shot (1-shot or 5-shot) NER task, the number of samples available in the divided validation set is too limited to prevent the overfitting problem.
> Thus, in our experiments, we do not divide the support set into a training set and a validation set, but use it as a whole for fine-tuning the model.
> To address the overfitting problem, we employ a loss-based early-stopping strategy, which is a commonly used approach in previous methods like CONTaiNER [1] during the fine-tuning step.
>
> In Appendix A.5, we provide specific implementation details.
> During the fine-tuning process of the span detection module and type classification module, we monitor the loss.
> If the loss continues to rise for β times (where β is a hyperparameter), we stop the fine-tuning process to mitigate the risk of overfitting. We will clarify this in the methodology section to ensure clarity and avoid any potential confusion.
>
> [1] Das et al., 2022. CONTaiNER: Few-Shot Named Entity Recognition via Contrastive Learning. ACL (1) 2022: 6338-6353.
>
> > Q6. The article argues that adding type names can be helpful for prototyping. In the result and analysis part of the article, only the influence on acc after adding the name is given, and the effect of the prototype is not intuitively seen. It seems that the article should give a vector diagram of the prototype to illustrate the correction or aggregation effect of the prototype.
>
> Thanks for the suggestion.
> We conducted visualization experiments as soon as we received comments from the reviewer, i.e., we visualized the distribution of prototypes and test samples without and with type names.
>
> The visualization results show that the entity prototypes of "ORG" and "MISC" types are obviously deviated from the corresponding actual sample centers, which is due to the scarcity of samples in the given support set. In contrast, the type-aware entity prototypes of the "ORG" and "MISC" types, which incorporate type name semantics, are able to alleviate the problem of prototype bias caused by the scarcity of samples.
>
> But, unfortunately, due to the format restriction of the rebuttal, we are unable to show the images here to the reviewer.
> We will add this visualization analysis in the paper to intuitively show the effect of type names on the prototype correction.
>
> > Q7. Is the model proposed in this paper an experiment based on the DecomposedMetaNER model? If so, is the classification of the training set, verification set, and test set the same as that of DecomposedMetaNER? Can you show more details of the processing?
>
>
> No, the experiment of our proposed model is not based on that of the DecomposedMetaNER [1] model.
> The only similarity between our approach and DecomposedMetaNER [1] is that it is also a decomposed framework, i.e., entity spans are extracted first and then entity types are categorized, which is common in NER [2,3].
>
> Besides, the division of the training set, validation set, and test set is the same as that of DecomposedMetaNER [1].
> Due to the space limit, we can only place a detailed description of the dataset in Appendix A.3. We apologize for the inconvenience.
>
> 1. We adopt the Few-NERD dataset made public by Ding et al. (2021) [4], which discloses the division of training/validation/test sets and few-shot sampling results (link: https://ningding97.github.io/fewnerd/).
> 2. For the Domain Transfer dataset, we use the division and sampling data made public by Das et al. (2022) [5] (link: https://github.com/psunlpgroup/CONTaiNER).
>
> Our TadNER and all the compared baselines, e.g., DecomposedMetaNER [1], conduct experiments using these two publicly available data divisions and samplings.
>
> [1] Ma et al., 2022. Decomposed Meta-Learning for Few-Shot Named Entity Recognition. ACL 2022, pages 1584–1596.\
> [2] Shen et al., 2021. Locate and Label: A Two-stage Identifier for Nested Named Entity Recognition. ACL 2021, pages 2782–2794.\
> [3] Zhang et al., 2022. Exploring Modular Task Decomposition in Cross-domain Named Entity Recognition. SIGIR 2022. Association for Computing Machinery, New York, NY, USA, 301–311.\
> [4] Ding et al., 2021. Few-NERD: A Few-shot Named Entity Recognition Dataset. ACL-IJCNLP 2021.\
> [5] Das et al., 2022. CONTaiNER: Few-Shot Named Entity Recognition via Contrastive Learning. ACL (1) 2022: 6338-6353.

---

### Meta-Review · Area_Chair_pbAD · 2023-09-15

**Recommendation:** 3

**Metareview:**

This article primarily enhances few-shot named entity recognition tasks by utilizing label information and span filtering to address two key challenges: the issue of over-detected false spans and the problem of inaccurate and unstable prototypes. The proposed method demonstrates strong performance in the realm of few-shot named entity recognition tasks, as substantiated by a comprehensive array of experiments. After serious discussion and consideration, we appreciate this solid work but think that the innovation appears somewhat limited. Furthermore, the explanation of filtering false spans is unclear, making it difficult to determine the positive impact of label information on the model and whether it filters out useful knowledge.

Overall, we appreciate both the efforts of reviewers and authors. This work is ready to publish, and the only concern is the novelty. We hope the authors can continue to improve the paper according to the comments and your response.

---

### Decision · Program_Chairs · 2023-10-07

**Decision:**

Accept-Findings

**Comment:**

This article primarily enhances few-shot named entity recognition tasks by utilizing label information and span filtering to address two key challenges: the issue of over-detected false spans and the problem of inaccurate and unstable prototypes. The proposed method demonstrates strong performance in the realm of few-shot named entity recognition tasks, as substantiated by a comprehensive array of experiments. After serious discussion and consideration, we appreciate this solid work but think that the innovation appears somewhat limited. Furthermore, the explanation of filtering false spans is unclear, making it difficult to determine the positive impact of label information on the model and whether it filters out useful knowledge.

Overall, we appreciate both the efforts of reviewers and authors. This work is ready to publish, and the only concern is the novelty. We hope the authors can continue to improve the paper according to the comments and your response.